# Epigenetic Regulation of Auxin-Induced Somatic Embryogenesis in Plants

**DOI:** 10.3390/ijms21072307

**Published:** 2020-03-26

**Authors:** Barbara Wójcikowska, Anna M. Wójcik, Małgorzata D. Gaj

**Affiliations:** Institute of Biology, Biotechnology and Environmental Protection, Faculty of Natural Sciences, University of Silesia in Katowice, Jagiellońska 28, 40-032 Katowice, Poland; anna.wojcik@us.edu.pl (A.M.W.); malgorzata.gaj@us.edu.pl (M.D.G.)

**Keywords:** auxin, DNA methylation, epigenetics, histone modifications, *MIRNA* genes, somatic embryogenesis

## Abstract

Somatic embryogenesis (SE) that is induced in plant explants in response to auxin treatment is closely associated with an extensive genetic reprogramming of the cell transcriptome. The significant modulation of the gene transcription profiles during SE induction results from the epigenetic factors that fine-tune the gene expression towards embryogenic development. Among these factors, microRNA molecules (miRNAs) contribute to the post-transcriptional regulation of gene expression. In the past few years, several miRNAs that regulate the SE-involved transcription factors (TFs) have been identified, and most of them were involved in the auxin-related processes, including auxin metabolism and signaling. In addition to miRNAs, chemical modifications of DNA and chromatin, in particular the methylation of DNA and histones and histone acetylation, have been shown to shape the SE transcriptomes. In response to auxin, these epigenetic modifications regulate the chromatin structure, and hence essentially contribute to the control of gene expression during SE induction. In this paper, we describe the current state of knowledge with regard to the SE epigenome. The complex interactions within and between the epigenetic factors, the key SE TFs that have been revealed, and the relationships between the SE epigenome and auxin-related processes such as auxin perception, metabolism, and signaling are highlighted.

## 1. Introduction

Since its discovery in the late 1950s, somatic embryogenesis (SE) is widely used for commercial plant micropropagation and transgenic plant production in plant biotechnology (reviewed in [1]). In addition to its practical value, SE provides a unique research system for studies on the molecular mechanisms that govern the developmental plasticity in plants [2]. The molecular pathways involved in the embryogenic response of in vitro-cultured plant explants are of particular interest in plant developmental biology because studies on SE contribute to the understanding of the regulatory mechanisms controlling toti- and pluripotency in plant somatic cells.

SE is induced by the transcriptomic reprogramming of the somatic plant cells that response to an induction signal, mostly after auxin treatment, and enter the embryogenic pathway of development and form embryo-like structures, the so-called somatic embryos. A great deal of the progress that has been achieved in the deciphering of the SE-regulatory network in recent years is attributed to the outcomes from studies on SE in Arabidopsis, a model in plant molecular genetics and genomics [3]. In Arabidopsis and numerous other plants, auxin treatment is the most efficient inducer of the embryogenic response in the in vitro cultured explants [4,5]. Similarly, auxin is a highly potent signaling molecule that controls almost every aspect of development in planta, including the promotion of cell division and elongation, root, leaf, flower, zygotic embryo, and fruit development [6,7,8]. A basic mechanism by which auxin triggers diverse developmental processes includes regulation of gene expression at the transcriptional level through the auxin signaling pathway and the core components of this pathway include the TRANSPORT INHIBITOR RESISTANT1/AUXIN SIGNALING F-BOX (TIR1/AFB) F-box proteins, the AUXIN/INDOLE-3-ACETIC ACID (Aux/IAA) transcriptional coregulators, and sequence-specific binding proteins called AUXIN RESPONSE FACTORs (ARFs) [9]. However, to understand the huge versatility of auxin-mediated developmental responses, crosstalk of auxin with epigenetic processes needs to be considered (Figure 1). In support of this approach, an increasing number of studies have suggested a close link between the auxin and epigenetic regulation of gene expression through miRNAs and chromatin modifications [10,11,12]. Consistently, the impact of miRNA regulation on auxin responses was evidenced, and specific miRNAs controlling the core elements of the auxin signaling pathway, including ARF, AUX/IAA, and TIR1/AFB receptors were identified [10,13].

miRNAs, which are products of the *MIRNA* genes, are small (19 to 24 nucleotides long), single-stranded non-coding RNA molecules that regulate both the state of the chromatin-associated with their targets and the availability of the encoded transcripts for protein translation [26]. Plant miRNAs have a high sequence specificity to their targets, and they tend to have fewer targets as compared with animal miRNAs [27]. Since their discovery in 2002 [28], plant miRNAs have been found to control many aspects of plant development, including their developmental plasticity and auxin responses [29,30]. During plant development, miRNAs preferentially target the genes that have a regulatory function, including those encoding transcription factors (TFs) and F-box proteins [31,32]. 

Auxin-mediated regulation of gene expression also involves the interplay of auxin with epigenetic modifications of DNA and histones, the processes of a pivotal role in controlling the development processes in animals and plants (reviewed by [33,34]). Methylation of DNA is the most intensively studied epigenetic mechanism that controls gene expression [35,36,37]. The methylation of plant DNA involves the addition of a methyl group to the carbon-5 of cytosine at the CpG, CpNpG, and CpNpN (N-any nucleotide except for G) sequences in DNA, which increases the content of 5-methyl cytosine (5mC) in the genome. The impact of DNA methylation on gene transcription has been widely evidenced, and the effect of 5mC depends on the gene region (promotor (P) vs. the gene body (GB)) that is to be methylated. Accordingly, the methylation of the gene promotor usually results in the repression of gene transcription, while an accumulation of 5mC in the GB is frequently associated with the constitutively expressed housekeeping genes [38]. Moreover, GB methylation seems to contribute to the regulation of the gene responses to internal or external cues [39]. The heterochromatin regions, imprinted genes, repetitive sequences, and transposons [40] display a high methylation level. In histone methylation, methyl groups are transferred to amino acid residues, mostly lysine, and the process is dynamically catalyzed by histone methylases and demethylases [41]. Depending on the target lysine and the degree of methylation, activation, or inactivation of genes could be induced [42]. The most frequent histone methylation marks, H3K27me3 and H3K4me3, are catalyzed by, respectively, the Polycomb-group (PcG) and Trithorax-group (TrxG) proteins, of documented roles in regulating plant developmental plasticity, including stem cell maintenance, cellular reprogramming, and plant responses to environmental cues (reviewed in [20], [43]). The PcG proteins are classified into two complexes, POLYCOMB REPRESSIVE COMPLEX 1 and 2 (PRC1 and PRC2), which cooperate to repress the genes via histone methylation during plant development [44]. The PRC2 of histone methyltransferase activity establishes the H3K27me3 markers to which the PRC1 complex binds and monoubiquitinates H2A histones (H2AK119ub1) in order to continually silence the transcription of a given genomic region [45]. A significant number (27.6%) of protein- and miRNA-coding genes in Arabidopsis have been found to be targeted by the H3K27me3 marker, which confirms the importance of the PRC-mediated gene silencing in plants [17]. Distinct PcG complexes exist in Arabidopsis, which differ in composition and function in plant development, and several protein members of the PRC2 and PRC1 complexes have been identified [20].

In histone acetylation, lysine residues on the N-terminal tails of histones undergo acetylation that results in weakening of the histone–histone and histone–DNA interactions, and in turn, the DNA accessibility to the chromatin-binding proteins is increased [46]. The dynamic changes in the state of histone acetylation is controlled by antagonistically acting enzymes, histone acetyltransferases (HATs), and histone deacetylases (HDACs) [47] and the interplay between the HAT and HDAC regulates developmental processes and stress responses (reviewed by [48,49]). The acetylation of histones is believed to promote open chromatin state and activate gene transcription [50].

To regulate gene expression in plant development, different epigenetic modifications, including DNA methylation and histone methylation and acetylation, extensively interplay following complex mutual interactions [51]. Insights into the methylome of numerous Arabidopsis mutants confirmed the complex regulatory pathways that mutually control DNA and histone methylation [52]. For example, the methylation of histone 3 at the K9 site was implicated in the regulation of DNA methylation in such a way that the SUVR2-conserved factor that is required for H3K9 methylation regulates the DOMAINS REARRANGED METHYLTRANSFERASE2 (DRM2) pathway of DNA methylation [52].

Numerous *TF* genes of auxin-related functions have been identified to control SE induction, and the complex regulatory interactions between the genetic regulators of SE have been evidenced [53]. However, to reveal the entire SE-regulatory network, epigenetic processes that orchestrate somatic cell transcriptome in response to the induction signal (mostly auxin) need to be explored. Here, we review the current state of knowledge about epigenetic regulation of SE induction and the evidence suggesting a role of miRNAs, DNA methylation, and histone modifications via acetylation and methylation, in SE induction, is reported.

## 2. Auxin-Related miRNAs Fine-Tune the Genetic Network that Controls SE

Extensive modulation of the *TF* genes in an embryogenic culture of Arabidopsis is accompanied by differential expression of numerous miRNAs, suggesting the function of miRNA-regulation in the embryogenic transition of plant somatic cells [54,55]. In support, the *dcl1* mutant, a defect in the *DICERLIKE1* (*DCL1*) gene with a key role in miRNA biogenesis, was shown to be unable for SE induction [56]. In addition to Arabidopsis, numerous miRNAs with a differential expression have been identified in embryogenic cultures of different gymnosperm, mono- and dicot plants, including economically important species [57,58,59,60,61,62,63,64,65,66,67,68,69,70,71,72,73,74,75,76,77]. A significant part of the SE-associated miRNAs comprises the auxin-related miRNAs whose target genes are engaged in auxin perception, signaling, and biosynthesis (Table 1). The auxin-related miRNAs frequently represented in embryogenic cultures involve miR165/166 and miR167 found in all of the analyzed SE systems: miR160, miR164, and miR390 identified in the majority of the SE-transcriptomes (77% to 92%), and miR393 expressed in some of the embryogenic cultures.

Transcriptomic datasets on *Zea mays* embryogenic cultures have shown that the expression pattern of individual auxin-related miRNAs is dependent on the genotype and the age of a culture [66,67,68,78]. The diversity of the miRNA expression profiles in embryogenic cultures, for example, the up- vs. downregulation of the same miRNA, could reflect the complexity of the miRNA-mediated regulatory pathways that fine-tune the somatic cell transcriptome towards embryogenic development in response to specific endo- and exogenous culture factors.

Numerous *MIRNA* genes, including members of the same gene family of SE-modulated expression, have been reported in transcriptomic analyses of different plant cultures [55,58,66,68,73,79]. The expression of different *MIRNA* gene family members is strictly regulated during plant development in an organ- and tissue-specific manner [80,81], and thus reporter line analyses could be important in identifying the specific *MIRNA* genes that contribute to SE induction. Accordingly, the expression of *MIR167c*, *MIR393a,b*, and *MIR396b* have been found to colocalize with the SE-induction sites of the explants, which supports the contribution of these genes to miRNA-mediated regulation of SE [56,82,83]. However, most of the reporter lines monitor the gene promoter activity rather than localize the corresponding gene transcript. Hence, a novel method of the whole-mount in situ hybridization (WISH) was recently proposed to analyze the spatiotemporal pattern of miRNAs in SE-induced tissue [84].

In addition, an analysis of the STTM (short target tandem mimic) and MIM (artificial miRNA target mimic) lines with a deregulated expression of *MIRNA* genes and miRNA targets [91,92] provided a powerful tool for identifying the miRNA function in plant development including SE induction. It was shown that a deregulated expression of *MIR160*, *MIR165*/*166*, *MIR167*, *MIR393*, and *MIR396* significantly affected the auxin accumulation and sensitivity to auxin in cultured explants of Arabidopsis [56,82,83,85].

The auxin-related miRNAs frequently target the fundamental components of the auxin signaling pathway, including the *ARF*s, which are involved in SE induction [27,93]. Accordingly, insights into embryogenic cultures of Arabidopsis and other plants indicated that miR160 could directly regulate the expression of *ARF10*, *ARF16*, and *ARF17*; miR167 seems to the control expression of *ARF6* and *ARF8*, while miR390 possibly downregulates the *ARF2*, *ARF3*, *ARF4* transcripts [55,63,64,68,81,85,86].

One possible outcome of the miRNA-mediated gene regulation in SE is the establishment of explant polarity, which is believed to affect the effectiveness of SE induction [94,95]. The miR165/166-mediated control of auxin biosynthesis during SE could account for the enhanced embryogenic response of the adaxial vs. abaxial sides of the immature zygotic embryo (IZE) cotyledon explants of Arabidopsis [22]. In this scenario, the miR165/166-mediated restriction of the PHABULOSA/PHAVOLUTA (PHB/PHV) transcripts to the adaxial cotyledon side results in a side-specific auxin accumulation due to the PHB/PHV-mediated stimulation of the LEAFY COTYLEDON2 (LEC2)-controlled pathway of auxin biosynthesis [85]. Support for the role of miR166-mediated regulation in the establishment of the embryogenic competency in the explant was provided in a recent analysis of an embryogenic culture of *Z. mays*. A unique pattern of miR166 (and several other small RNAs, miR156, miR164, and tasiARFs) was found associated with a specific SE-responsive developmental stage of an explant, the IZE [96]. In leaf development, miR165/166 is under the control of ARF3 and ARF4, which are polarly distributed as a result of the miR390-dependent tasiARFs that move intracellularly from the adaxial to the abaxial side of a leaf [97,98] (reviewed in [99]). Although whether or not miR390 controls ARF3/4 in SE-induced explants needs verification, the miR390-ARF3/4 module operates in the auxin-signaling pathway to regulate lateral root development [100] of genetic convergence into auxin-induced tissue dedifferentiation and callus formation in vitro [101].

In addition to targeting PHB/PHV, miR165/166 could also impact SE by repressing the WUSCHEL-RELATED HOMEOBOX5 (WOX5) TF, which has a reported role in the formation of the root apical meristem (RAM) in the somatic embryos of Arabidopsis [102,103]. Although miR165/166, together with miR160, is known to target *WOX5* in order to regulate root development in zygotic embryos and seedlings [104,105], the contribution of the similar regulatory module to SE remains to be revealed.

Auxin-related miRNAs could contribute to SE induction by regulating the metabolic pathways connected with other hormones, including ethylene. Accordingly, GhmiR157a modulates the free auxin level in *Gossypium hirsutum* by controlling the ethylene content, which triggers the callus initiation and cellular dedifferentiation that are associated with the embryogenic transition of somatic cells [106].

The miRNA-mediated SE-regulatory network also involves miR164, which seems to control the CUP-SHAPED COTYLEDON1 (*CUC1*) and *CUC2* genes in SE [55]. The *CUC* genes encode the TFs of the NAC family and control the establishment of the shoot apical meristem in the zygotic embryos of Arabidopsis [107]. It has also been found that TF TEOSINTE BRANCHED1, CYCLOIDEA and PCF (TCP3) can regulate the expression of miR164 during shoot meristem development [108].

The complex miRNA-mediated regulatory network of SE induction considers feedback loops between the miRNA and the targets. Accordingly, miR390-targeted ARF5/MONOPTEROS (MP) was recently found to bind auxin-responsive elements (AuxRE) in the promoter of the *MIR390a* gene to control the *MIR390* expression in the root meristem [109]. Although both ARF5/MP and miR390 have been suggested to control SE induction in Arabidopsis, their regulatory interactions in this process require experimental validation [55,93].

A regulatory feedback loop also operates within the miR393-TAAR (clade of TIR1/AFB auxin receptors) module with a central role in regulating the auxin responses during plant development via auxin-regulated degradation of Aux/IAA transcriptional repressors [57,110,111]. Similar to its function in planta, miR393 has been shown to modulate the expression of the *TIR1* and *AFB2* auxin receptors to control the sensitivity of explants to auxin treatment and homeostasis of auxin signaling during the embryogenic transition in Arabidopsis [56]. In addition to Arabidopsis, the expression of miR393 and its *TIR1* and *AFB* targets has also been found in the SE-transcriptomes of other species, including *Citrus sinensis* [59], *Dimocarpus longan* [62], *G. hirsutum* [65], *Z. mays* [67], and *Triticum aestivum* [69], which suggests that there is a common miR393 function in auxin perception during SE induction.

The complex interactions of different miRNA-controlled modules could be expected in the auxin-induced SE. The miR165/166 together with miR160 have been postulated to contribute to the LEC2-controlled pathway of SE induction by targeting *PHB/PHV* and *ARF10/ARF16*, respectively [85]. A similar regulatory interaction between miR160 and miR165/166 was also recently found to control the auxin-mediated leaf development and drought tolerance in Arabidopsis [112]. The miRNA controlled ARFs of overlapping expression patterns could interact genetically to fine-tune the auxin responses. As an example, the miR160-repressed ARF17 interacts with the miR167-controlled ARF6 and ARF8 in the complex network controlling the adventitious root initiation [113]. Although the contribution of miR167-ARF8 to SE induction needs experimental verification, the role of this module in adjusting the cellular free auxin level in an *O. sativa* cell culture was reported [114].

Recently, the GROWTH-REGULATING FACTORs (GRF1, GRF4, GRF7, GRF8, and GRF9) and PLETHORA (PLT1 and PLT2) TFs have been postulated to be among targets of miR396 in an Arabidopsis embryogenic culture. The complex regulatory interactions within the miR396-GRF-PLT module that involve the negative and positive regulation of GRFs and PLT by miR396, respectively, could be assumed to control the embryogenic transition of the somatic plant cells [83]. Similarly, a miR396 function has been associated with regulating the developmental switch in the root stem cells in planta [115].

Little is known about the regulators of the SE-associated auxin-related miRNAs and among the candidates is *FUSCA3* (*FUS3*), which is structurally and functionally related to *LEC2*, and was suggested to control the *MIR156, MIR160, MIR166*, and *MIR396* genes in an embryogenic culture of Arabidopsis [116]. In addition, another key regulator of the embryogenic transition, the AGAMOUS LIKE15 (AGL15) TF, which interacts with *LEC2* and *FUS3* in the auxin-mediated pathway of SE induction (reviewed in [117]) has been postulated to control auxin signaling in both Arabidopsis and soybean via the direct stimulation of a miR167a-encoding gene [118].

It is noteworthy that in addition to controlling the SE-induction pathway, auxin-related miRNAs also appear to have a crucial impact on the embryogenic competency of an explant. A recent analysis of SE in maize showed that the specific developmental stage of an IZE explant recommended for efficient SE induction displayed a unique pattern of several auxin-related miRNAs, including miR156, miR164, miR166, and miR393 [97]. Similarly, in Arabidopsis, an efficient SE response is associated with a strictly defined developmental stage of an IZE explant, and this significantly limits the establishment of efficient in vitro regeneration systems for biotechnology [3]. Thus, defining the miRNA-status of the SE-responsive explant stage could contribute to identifying new factors that determine the embryogenic capacity of tissue and, as a result, improve the plant regeneration protocols for in vitro-recalcitrant genotypes.

## 3. Epigenetic Modifications and Auxin Responses Interact Closely to Control the Embryogenic Transition

### 3.1. DNA Methylation

In a pioneering work on DNA methylation in SE induction, a treatment with auxins but not cytokinins was found to increase the level of global DNA methylation in embryogenic cell cultures of *Daucus carota* [119]. The impact of 2,4-D on DNA methylation has also been confirmed in embryogenic cultures of *Picea omorika* [120], *Malus xiaojinensis* [121], and *Glycine max* [122]. The changes in global 5mC content in the auxin-induced embryogenic cultures of different plants have been associated with a differential gene expression [123], and an extensive modulation of the genes encoding both the methylases and demethylases of DNA has been attributed to the auxin-induced SE-transcriptomes of different plants, i.e., *Populus trichocarpa*, Arabidopsis, *G. hirsutum*, and *G. max* [122,124,125,126]. In Arabidopsis, DOMAINS REARRANGED METHYLTRANSFERASE1-2 (DRM1-2) are required for de novo methylation via RNA-directed DNA methylation (RdDM) pathway, while METHYLTRANSFERASE1 (MET1) and CHROMOMETHYLASE1-3 (CMT1-3) maintain the methylation pattern during DNA replication. Similar to the early stages of zygotic embryo formation [127], *MET1* and *CMT3* had a significantly higher expression level than *DRM1* and *DRM2* during SE in Arabidopsis [24]. Several reports have implied that auxin (2,4-D) treatment could regulate genes encoding (de)methylases differently, and consistently, an auxin medium increased the expression of the *DRM*s and *CMT3* methylases and downregulated the expression of the (*REPRESSOR OF SILENCING1*) *ROS1, DEMETER* (*DME*), and *DEMETER-LIKE PROTEIN2* (*DML2*) demethylases in embryogenic cultures of *D. carota* and Arabidopsis [24,128,129]. The presence of the auxin responsive AuxRE motif, in the *CMT3* promoter, suggests a direct impact of auxin on gene transcription. In contrast, auxin seems to indirectly control the AuxRE-less promotor of the *MET1* gene, and the underlying mechanism remains to be revealed.

Evidence for a link between DNA methylation and auxin signaling provides a significantly modified embryogenic response of Arabidopsis mutants impaired in methylases including *met1* and *drm* [24]. A mutation in the *MET1* gene, which significantly decreased the global level of DNA methylation [130], promoted faster in vitro shoot regeneration, due to early activation of auxin-induced *WUSCHEL* (*WUS*) [131]. In addition, the *met1* mutation caused an abnormal auxin gradient in embryos and deregulation of the PIN-FORMED1 (PIN1) auxin efflux carrier engaged in polar auxin transport in plant development, including SE [132,133]. The role of DNA methylation in the regulation of the PIN-like proteins during somatic embryo development in *Araucaria angustifolia* has been suggested [134]. The expression of the DNA methylase genes in SE cultures, together with the significantly modified embryogenic response of the DNA methylase mutants, implies that both the maintenance and de novo pathways of DNA methylation are engaged in regulating SE induction. However, the complexity of the interactions controlling the balance between DNA replication de novo and the maintenance of DNA methylation and DNA demethylation [135] results in a lack of a direct relationship between the activity of the DNA (de)methylases and the global 5mC status of a genome [136,137].

In the majority of reports, an inverse relationship has been observed between the embryogenic competence of explants/culture and the DNA methylation level. Accordingly, the hypomethylation of DNA seems to be associated with the early stages of embryogenic induction, and the embryogenic cultures have a lower level of 5mC than the non-embryogenic cultures (reviewed in [25]). However, the hypomethylation of DNA is not specific to the embryogenic induction and seems to be related to a general process of tissue dedifferentiation because both the embryogenic culture and non-embryogenic calli of Arabidopsis had a decreased 5mC content [24,138]. Conversely, the increased global level of DNA methylation could be associated with SE induction, as was reported in *Eleuterococcus senticosus* [139]. Because of the variety of factors, including the stress conditions, organ/tissue type and genotype impacting the methylation status of a genome [40], diversity in the SE-associated 5mC patterns could be expected between different embryogenic cultures.

The contrasting effects of a demethylation agent, 5-azacitidine (5-AzaC), treatment on the embryogenic response of plant tissue offer further evidence that a complex set of exo- and endogenous culture factors rather than the induction of embryonic development per se appear to shape the global DNA methylation level in in vitro cultured tissue. Accordingly, 5-AzaC treatment has been recommended for improving the embryogenic capacity of cultures derived from poorly responding plant explants and the genotypes of in vitro recalcitrant plant species [25] including the recovery of the SE potential in aged cultures of *Theobroma cacao* [140]. However, the 5-AzaC treated explants of Arabidopsis showed a significantly reduced SE capacity [24]. The factors that could modify the effect of 5-AzaC on in vitro cultured tissue involve hormonal treatment, and 2,4-D was reported to modulate the effect of 5-AzaC on the embryogenic response in *Acca sellowiana* in a concentration-dependent manner [141].

In contrast to the global level of 5mC, which seems to be unspecific to embryogenic vs. non-embryogenic tissue [24], an analysis of DNA methylation in response to SE induction signal at the gene level could be more conclusive for revealing the interactions between the DNA methylation, auxin, and gene expression that promote SE. However, studies on the methylation of the specific genes associated with in vitro-induced plant morphogenesis, including SE, are rather limited. A decreased 5mC level has been found in the promotors of *LEAFY COTYLEDON1* (*LEC1*) and *WUS*, respectively, in an embryogenic culture of *D. carota* [142] and during shoot organogenesis in Arabidopsis [131,143]. Similarly, a decreased methylation level has been reported in the *SOMATIC EMBRYOGENESIS RECEPTOR-LIKE KINASE* (*SERK*), *LEC2*, and *WUS* genes in the embryogenic calli of *Boesenbergia rotunda* [144]. Hypomethylation at the CHH sites in a promoter has been associated with the activation of some hormone related and *WOX* genes in *G. hirsutum* SE [126].

To summarize, the hypomethylation of the regulatory genes, including *TF*s, seems to be a typical response of de-differentiated somatic cells; however, more genes need to be analyzed to verify this assumption. Insight into the methylation of the specific sequences within genes is of particular interest given that DNA methylation is a site specifically regulated [52] and the methylation of the promoter and coding sequences seems to have a significantly different function [40,145]. Consistently, the decreased vs. increased number of genes with methylated P and GB sequences, respectively, have been found during an in vitro dedifferentiation and regeneration of *P. trichocarpa* [124].

In addition to auxin (2,4-D) treatment, the in vitro imposed abiotic stress could modulate plant somatic cell methylome given that differentiated genome methylation has been postulated as providing the adaptive mechanism to different stresses in plants [40,146]. This finding, together with dual, stress-, and auxin-like responses induced by 2,4-D make identifying the gene-specific DNA methylation patterns that promote auxin-induced SE challenging.

### 3.2. Histone Methylation

Several reports have indicated the interplay between DNA and histone methylation in the chromatin remodeling to be associated with the in vitro-induced developmental processes, including SE in *G. hirsutum* [126]. A decrease in the 5mC level and histone methylation at the H3K9me2 and H3K9me3 markers has been attributed to the positive regulation of gene expression during plant cell dedifferentiation [147,148]. Similarly, a decrease in the DNA methylation level has been associated with a decrease of H3K9me2 and H3K27me3 during SE in *Coffea canephora* [149].

The involvement of PRC2-mediated histone methylation in the repression of the embryo maturation programs during the vegetative development in Arabidopsis has been indicated [18]. The cell fate can be reset, and SE could be induced in PRC2-depleted tissues treated with hormone [18]. The role of PRC2 in the reprogramming of somatic cells has also been suggested for the callus and somatic embryo production that is associated with the loss-of-function mutations in the genes encoding the proteins of the PRC2 complex, CURLY LEAF (CLF), SWINGER (SWN), VERNALIZATION2 (VRN2), and EMBRYONIC FLOWER2 (EMF2) [19]. Consistent with a pluripotency-related function, PRC2 has been suggested to target the TFs, including LEC1, LEC2, FUS3, AGL15, PLT, and WOXs, which have regulatory roles in somatic cell differentiation and SE induction in different plants including Arabidopsis, *C. canephora*, and *Medicago truncatula* [149,150,151,152]. Within the PRC2-repressed targets, the genes involved in biosynthesis, transport, perception, and signal transduction of auxin have been observed [17]. Accordingly, PRC2 has been found to directly silence the *TRYPTOPHAN AMINOTRANSFERASE RELATED1* (*TAR1*) and *YUCCA10* (*YUC10*) genes involved in the indole-3-acetaldoxime-mediated auxin biosynthesis pathway with a role in SE induction in Arabidopsis [153,154]. The role of PRC2 in controlling auxin responses in vitro also implies a significantly reduced level of H3K27me3 in the chromatin-associated with the auxin-related genes, *INDOLE-3-ACETIC ACID INDUCIBLE2* (*IAA2*)*, GH3.2, NITRILASE* (*NIT2*)*, YUC4, IAA CARBOXYLMETHYLTRANSFERASE1* (*IAMT1*), and *PIN1* during callus production [155]. The PRC2-mediated repression of the PIN1 auxin efflux carrier has been indicated to downregulate the auxin maxima in the lateral roots of Arabidopsis [156]. Because of the similarity of root development to the callus and plant regeneration pathway [101], including the role of *YUC4* in both SE induction and lateral root development [153,157], the PRC2 regulation of auxin transport during the embryogenic transition of somatic cells could be assumed.

The PRC1 complex in Arabidopsis consists of five proteins, AtRINGa/b and AtBMI1a-c, and the *Atbmi1a Atbmi1b* and *Atring1a Atring1b* double mutant seedlings have shown a spontaneous callus and somatic embryo formation [14,15]. Consistently, PRC1 was found to negatively control the TFs of a regulatory role in embryogenic development, including ABSCISIC ACID INSENSITIVE3 (ABI3), AGL15, BABY BOOM (BBM), FUS3, LEC1, LEC2, WOXs and the auxin transporters of the PIN family [15].

In line with the negative impact of PRC1 on SE, a decreased versus an increased expression of *PRC1* genes (*RING1*, *BMI1*, *LIKE HETEROCHROMATIN PROTEIN1 LHP1*, *EMBRYONIC FLOWER1 EMF1*, and *VERNALIZATION1 VRN1*) has been found in the embryogenic vs. non-embryogenic genotypes of *M. truncatula* [16], respectively. In control of the SE-genes, the PRC1 complex interacts with the VP1/ABI3-LIKE1-3 (VAL1-3) proteins, and a high embryogenic potential of the *val1 val2* mutant is accompanied by the activation of the embryonic genes *LEC1*, *LEC2, FUS3*, and *ABI3* [158,159].

Unlike PRC1 and PRC2, the TrxG complexes are frequently associated with actively transcribed genes [160]. Interestingly, depending on the plant’s developmental stage that is controlled, the proteins of the PRC and TrxG complexes can interact in an antagonistic vs. cooperative manner. For example, to prevent any precocious seed gene expression after germination, the ARABIDOPSIS HOMOLOG OF TRITHORAX1 (ATX1), ULTRAPETALA1 (ULT1) of the TrxG, and EMF1 of the PRC1/2 work together to maintain chromatin integrity [21]. The TrxG proteins control gene expression in plant development via multiple mechanisms that include modifying the chromatin structure via the deposition of the H3K4me2/3 marker [161]. The TrxG factors, the ATX1 of H3K4me3 methyltransferase activity, and ULT1 interact physically to reduce the H3K27me3 marker in the targeted genes, including the SE-regulators, *LEC1, LEC2, FUS3*, and *ABI3* [20]. The direct binding of both ATX1 and ULT1 to the seed gene loci, *ABI3* and *LEC2*, was observed [21].

The essential regulatory function of TrxG/PRC complexes in the auxin-mediated processes implied a vast number of the auxin-related genes that showed a misregulation of expression in the mutants impaired in the genes encoding the TrxG and PRC proteins [21]. BRAHMA (BRM) of the TrxG complex has been found to directly activate the auxin efflux carriers (PINs) in part by antagonizing the H3K27me3-associated chromatin repression that is mediated by the PcG proteins [162]. The proteins of TrxG also cooperate with the components of the auxin signaling pathway, and in response to auxin, ARF5 recruits the chromatin remodeling complexes (BRM and SPLAYED) to the target loci in order to overcome the repressed chromatin state [163].

The histone methylation-mediated control of gene expression is also orchestrated by other proteins that act antagonistically to PcG. Among them, the members of the JUMONJI C (JmjC)-domain-containing-protein family have been identified. These proteins regulate various genes, including the PIN auxin carriers and auxin biosynthesis *YUC3* gene via the direct histone demethylation [164,165]. Further studies are needed to determine whether the (JmjC)-domain-containing-proteins also control the auxin-related SE response.

### 3.3. Histone Acetylation

A model of the histone acetylation contribution to auxin signaling considers interactions between the auxin-responsive elements (AuxRE) and the HAT acetylases [166]. Consistently, AuxREs have been identified in the promoters of the majority of the *TF* genes of upregulated expression during SE induced in Arabidopsis with the HDAC-inhibitor, trichostatin A (TSA) [22]. Evidence of the contribution of the (de)acetylation of histones to the in vitro-induced embryogenic transition also provides the differential expression of several members of the *HDAC* and *HAT* gene families in the SE transcriptomes of plants [125,167,168,169]. Two of the deacetylases, HDAC6 and HDAC19, seem to be of particular importance in SE induction as the *hdac6/hdac19* mutant plants showed somatic embryo development on their leaves [23]. The HDAC6 and HDAC19 control the genes expression in zygotic embryos, and the HDAC19-mediated repression of *LEC1* and *LEC2* via the deacetylation of histone H3 was reported [170,171,172].

Experimental evidence about the histone acetylation-mediated control of in vitro-induced plant morphogenesis, including SE are still limited; however, several reports have indirectly implied such a control. An increased level of H3 and H4 acetylation associated with the expression of the histone acetyltransferase genes has been demonstrated during the microspore embryogenesis of *Brassica napus* [167] and an increased level of a specific histone acetylation marker (H3K9 and 14) accompanied the chromatin decondensation and transcriptional activation of the genes during the dedifferentiation of *Nicotiana tabacum* protoplasts [173]. Experiments with TSA also suggest that histone acetylation plays a role in SE control [174], and the TSA-mediated inhibition of HDACs results in histone hyperacetylation and conformational changes of the chromatin-associated with enhanced gene expression [175]. Consistently, an increase in the H3K9/K14Ac and H4K5Ac epigenetic markers was reported in the TSA-treated seedlings of Arabidopsis [176,177]. In support of a role of histone acetylation in controlling SE induction, TSA treatment promoted the development of embryogenic structures on the seedlings and in vitro-cultured explants of Arabidopsis and conifers [22,23,178,179] and the beneficial effects of TSA on microspore cultures of *T. aestivum* [180] and *B. napus* [181] were reported. Nonetheless, details of the TSA-promoted regulatory interactions that provoke an embryogenic response in somatic plant cells has not yet been clarified. Some clues about the TSA-mediated SE induction mechanism indicated an auxin accumulation accompanied by the upregulation of the auxin-related SE-regulators, including *LEC1*, *LEC2*, *BBM*, and *PHB* in a TSA-induced embryogenic culture of Arabidopsis [22]. In addition to Arabidopsis, the TSA-induced upregulation of *LEC1* and *LEC2* has also been reported in microspore culture of *B. napus* [181], and the transcripts of *LEC1*-type *HAP3* were accumulated in the TSA-treated IZEs of *Picea abies* [178]. Consistent with the TSA-upregulated expression of *LEC1* and *LEC2*, the histone deacetylase complex was implicated in the repression of the *LECs* and other genes of the LAFL (LEC1, ABI3, FUS3, LEC2) network, which controls seed development [182]. Interestingly, insight into the *TF* genes expressed in plant embryogenic cultures suggests that histone acetylation could preferentially activate the expression of the auxin-responsive *TF* genes [22,23,178,181]. In addition to *TFs*, the genes that were upregulated in the TSA-induced SE included the *YUC* (*YUC1* and *YUC10*) genes of the auxin biosynthesis pathway [22] and the genes associated with the metabolism (*GH3*), transport (*PIN1, PIN3*, and *PIN7*) and signaling (*AFB3*) of auxin [181]. Similar to plants, TSA was found to derepress the transcription of the *TF* genes controlling the reprogramming of mammal somatic cells into embryonic stem cells [183]. The gene-specific effects of TSA were postulated, and in support, the non-stochastic impact of TSA treatment on the transcriptomes of diverse organisms, including humans, other mammals, and Arabidopsis, was reported [176,183,184].

The detailed mechanisms by which the TSA-modulated histone acetylation status of chromatin affects the SE-involved genes remains elusive. In a possible scenario, the SE-induction signal (provided by TSA or 2,4-D treatment) promotes histone acetylation and activates the genes encoding the pioneer TF of a crucial function in the embryogenic transition, including LEC1 [185]. LEC1 is assumed to act as a pioneer TF that promotes the post-translational modifications of the core histones involved in regulating gene expression [186]. It is noteworthy that among the direct targets of LEC1, the *TFs* encoding the LAFL-group regulators of the SE-involved auxin biosynthesis pathway were identified [117,172].

The current model of auxin-regulated gene expression implies a relationship between the auxin- and TSA-responsiveness of genes (reviewed in [9]). Consistent with this model, some genetic convergence of the TSA- and 2,4-D-induced embryogenic development was revealed in Arabidopsis [22]. Similar to in planta development, auxins could promote SE-regulatory genes via the histone acetylation-mediated derepression of gene transcription. Moreover, the capacity of TSA for gene derepression seems to be stronger than that of 2,4-D, as TSA is exclusively able to activate the SE-regulatory genes in the post-germinated tissue of Arabidopsis [23]. Thus, although TSA seems to target the auxin-responsive genes preferentially, the effects of TSA vs. 2,4-D on the gene expression level could differ to some extent. In support of this, the downregulation vs. the upregulation of *ARF10* and *ARF17* during TSA- vs. 2,4-D-induced SE, respectively, was reported in Arabidopsis [22,93]. The targets of histone acetylation also involve the *MIRNA* genes, including those encoding auxin-related miRNA [187,188,189,190]. TSA treatment affected the accumulation of miRNAs, and GCN5-dependent H3K14 acetylation was required for the transcriptional regulation of the auxin-related miR164, miR165, and miR167, which have a regulatory function in SE induction in Arabidopsis [55,187,188].

It is also possible that the TSA-induced SE mechanism could comprise other TSA-affected epigenetic processes, including the methylation of DNA and histones [191,192] and other TSA-targeted modifications which are unrelated to gene transcription. Hence, understanding the intricate mechanisms that control the derepression of the embryogenic potential in somatic plant cells requires deciphering the complex interplay between histone and non-histone acetylation and the other epigenetic processes that regulate gene expression.

## 4. Concluding Remarks and Perspectives

The extensive studies on embryogenic cultures of plants, primarily on the model plant Arabidopsis, have revealed the complex interactions between the regulatory genes in which the auxin-related epigenetic processes have been found to play a central role (Figure 2). Although the SE-regulatory network has some similarities to the genetic and epigenetic control of zygotic embryogenesis, recent RNAseq data of an embryogenic culture of Arabidopsis unexpectedly revealed that the SE transcriptome differs from the early transcriptome of a zygotic embryo and resembles the gene expression pattern of germinating seeds [193]. These results shed new light on the molecular events that control SE induction and indicate that identifying the core, canonical components of the common molecular regulatory pathway governing embryonic development in zygotic and somatic plant cells could be extremely challenging. Moreover, due to the variety of endo- and exogenous factors that orchestrate the cell transcriptomes, a high degree of diversity could be expected between the SE transcriptomes. Thus, searching for the epigenetic events that determine the SE-response in various plants and explant types could be of particular interest in further investigations of the regulatory determinants of the embryogenic response. Methylation-related processes seem to be of particular interest as the genes involved in the methylation of DNA and histones have recently been found to be predominantly active during the epigenetic reprogramming in early zygotic embryogenesis [194].

## Figures and Tables

**Figure 1 ijms-21-02307-f001:**
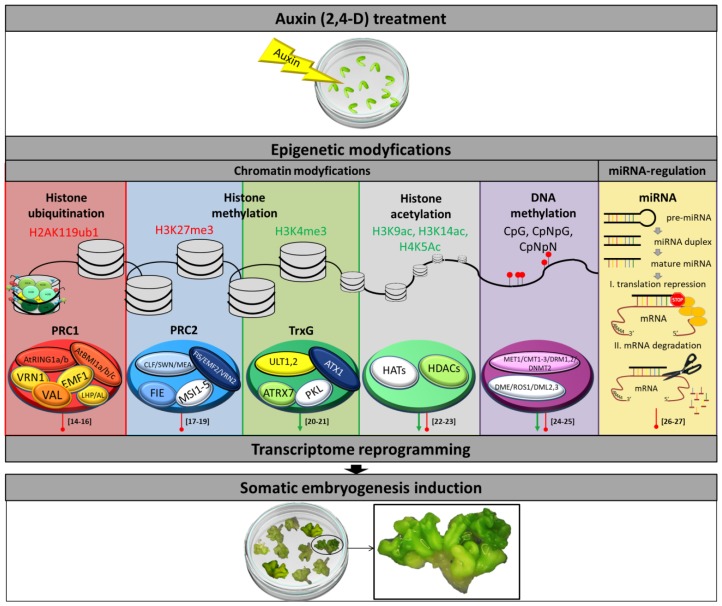
An overview of the epigenetic processes, including chromatin modifications and miRNA-mediated gene regulation that control auxin-induced embryogenic response of explant cells. Modifications of chromatin, including histone methylation, acetylation, and ubiquitination together with DNA methylation, control the transcriptome of explant cells in response to auxin treatment. The epigenetic complexes controlling the SE-transcriptome involve both the repressors (PRC1, PRC2, HDACs, DRM1-2, MET1, CMT1-3, and DNMT2) and activators (TrxG, HATs, DME, ROS1, and DML2-3) of gene expression. The specific epigenetic marks that activate and repress gene transcription are indicated in green and red, respectively. Arrows and circle-shaped ends show the activation or repression of gene expression, respectively [14,15,16,17,18,19,20,21,22,23,24,25,26,27]. (2,4-D) 2,4-dichlorophenoxyacetic acid; (AL) ALFIN-LIKE; (AtBMI1a/b/c) B LYMPHOMA Mo-MLV INSERTION REGION 1 HOMOLOGa/b/c; (AtRING1a/b) RING FINGER PROTEIN1a/b; (ATRX7) Arabidopsis TRITHORAX-RELATED7; (ATX1) ARABIDOPSIS HOMOLOG OF TRITHORAX1; (CLF) CURLY LEAF; (CMT1-3) CHROMOMETHYLASE1-3; (DME) DEMETER; (DML2,3) DEMETER-LIKE PROTEIN2,3; (DNMT2) DNA NUCLEOTIDE METHYLTRANSFERASE2; (DRM1,2) DOMAINS REARRANGED METHYLTRANSFERASE1,2; (EMF1,2) EMBRYONIC FLOWER1,2; (FIE) FERTILIZATION-INDEPENDENT ENDOSPERM; (FIS) FERTILIZATION INDEPENDENT SEED; (HATs) HISTONE ACETYLTRANSFERASEs; (HDACs) HISTONE DEACETYLASEs; (LHP) LIKE HETEROCHROMATIN PROTEIN; (MEA) MEDEA; (MET1) METHYLTRANSFERASE1; (miRNA) microRNA; (MSI1-5) MULTICOPY SUPPRESSOR OF IRA1-5; (PKL) PICKLE; (PRC1,2) POLYCOMB REPRESSIVE COMPLEX1,2; (ROS1) REPRESSOR OF SILENCING1; (SWN) SWINGER; (TrxG) Trithorax-group; (ULT1,2) ULTRAPETALA1,2; (VAL) VP1/ABI3-LIKE1-3; (VRN1,2) VERNALIZATION1,2.

**Figure 2 ijms-21-02307-f002:**
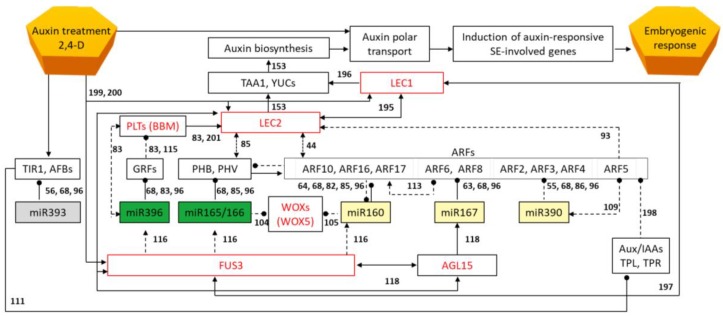
The regulatory network controlling SE induction in which the auxin-related epigenetic processes play a central role. A number of auxin-related miRNAs orchestrate the explant cell transcriptome towards embryogenic transition through regulation of auxin perception (miR393), biosynthesis (miR396 and miR165/166), and signaling (miR160, mi167, and miR390). The SE-involved miRNAs control embryogenic induction by targeting the *TF* genes with a key regulatory function in embryonic development including the *LEC1, LEC2*, and *FUS3* of the *LEC* group; *BBM* (*PLTs*); *AGL15* and *WOXs* (*WOX5*) genes; and a number of *ARF*s involved in the auxin-signaling pathway. The miRNAs controlling auxin perception, biosynthesis, and signaling are indicated by the grey, green and yellow boxes, respectively. The key *TF*s are marked in red including the genes of the *LEC* group, which are distinguished by red-framed boxes. Arrows and circle-shaped ends show the activation or repression of gene expression, respectively. Solid vs. dashed lines refer to the experimentally validated and suggested regulatory interactions that operate during SE, respectively [195,196,197,198,199,200,201]. (2,4-D) 2,4-dichlorophenoxyacetic acid; (AFBs) AUXIN F-BOX PROTEINs; (AGL15) AGAMOUS-LIKE15; (ARF) AUXIN RESPONSE FACTOR; (Aux/IAA) AUXIN/INDOLE-3-ACETIC ACID; (BBM) BABY BOOM; (FUS3) FUSCA3; (GRFs) GROWTH-REGULATING FACTORs; (LEC1,2) LEAFY COTYLEDON1,2; (miR) microRNA; (PHB) PHABULOSA; (PHV) PHAVOLUTA; (PLT) PLETHORA; (TAA1) TRYPTOPHAN AMINOTRANSFERASE OF ARABIDOPSIS1; (TIR1) TRANSPORT INHIBITOR1; (TPL) TOPLESS; (TPR) TOPLESS RELATED; (WOX) WUSCHEL-RELATED HOMEOBOX; (WUS) WUSCHEL; (YUC) YUCCA.

**Table 1 ijms-21-02307-t001:** The miRNA molecules involved in auxin perception, signaling, and biosynthesis expressed in the SE transcriptomes of different plants [85,86,87,88,89,90]. x, transcriptomic analysis; ●, functional analysis.

		AUXIN PERCEPTION	AUXIN SIGNALING			AUXIN BIOSYNTHESIS		OTHER
miRNA Name		miR393	miR160	miR167	miR390	miR165/166	miR396	miR164
**Target genes**		*TIR1, AFB1, AFB2, AFB3*	*ARF10, ARF16, ARF17*	*ARF6, ARF8*	*TAS3, ARF2, ARF3, ARF4, ARF5*	*PHB, PHV, HDZ31, HDZ32, HDZ33*	*GRFs*	*NAC1, CUC1, CUC2*
**SPECIES**	***A. thaliana***	● x	● x	● x	x	● x	● x	x
	***Z. mays***	x	x	x	x	x	x	x
	***C. sinensis***	x	x	x	x	x	x	x
	***D. longan***	x	x	x	x	x		x
	***G. hirsutum***	x	x	x	x	x	x	x
	***L. leptolepis***		x	x	x	● x	x	x
	***L. pomilum***		x	x	x	x	x	x
	***L. tulipifera×*** ***L. chinense***		x	x	x	x	x	x
	***T. aestivum***	x		x		x	x	x
	***P. balfouriana***		x	x	x	x	x	x
	***O. sativa***			x		x		x
	***C. nucifera***			x		x		x
	***P. pinaster***		x	x	x	x	x	
**References**		[55,61,63,65,70,71,72,73,88]	[55,61,62,63,64,65,68,71,73,75,78,79,82,85]	[55,56,57,58,61,62,63,64,65,67,71,74,75,78,82,85,89,90]	[55,57,58,61,62,63,65,66,71,75,78,82,85]	[55,56,57,58,59,61,62,63,65,70,71,73,74,75,78,82,85,86,87,89,90]	[55,59,61,65,67,69,70,71,72,75,78,79,83]	[55,57,58,61,62,63,69,70,71,73,74,78,79,85,89]

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
