# Peer review of "Epigenetic Regulation of Auxin-Induced Somatic Embryogenesis in Plants"

_ijms, 2020, doi:10.3390/ijms21072307_

Round 1
Reviewer 1 Report
This review by Wójcikowska et al. nicely summarizes the regulatory roles of miRNAs and chemical modifications in auxin-induced somatic embryogenesis in plants. It seems that most important studies in this field were covered, but the way the review was written and presented can still be improved. I strongly suggest that the authors ask one or two biologists who are native speakers to read through the text carefully and to help improve the sentence structures and the readability of the manuscript.
My general feeling is that many sentences are too long and quite wordy, sometimes with redundant words, superfluous phrases, syntactical errors, and overuse or incorrect use of clauses/parentheticals/commas that can disrupt sentence flow and result in nonsensical parts. It should be noted that some phrases/words occur multiple times, have relatively little meaning, and often make sentences more difficult to comprehend, such as "in particular" (6 times in the text) and "relevant"/"relevantly" (10 times). Below are some suggestions:
Lines 10-11: In this example, "that accompany" can be changed to "during", which would reduce the number of "that" relative clauses in the sentence.
Line 12-15:
- Is "among these factors," necessary?
- What does "significantly and relevantly," mean?
- There are two relative clauses: "that have an ..." and "that control SE". Could they be reduced?
- The second part of this sentence has a very heavy subject "several miRNAs that have an essential function in regulating the transcription factors (TFs) that control SE" and a short body "have been identified in the past few years". I would rather have a shorter subject and see the finite verb earlier.
Line 15-17: The "Relevant ...," part doesn't seem to add any information here. The sentence says "auxin-related" twice and "auxin" thrice, which is quite redundant. I would suggest to change this sentence to "Most of these miRNAs were found to be involved in auxin metabolism and signaling."
Lines 67-72: It would be better to make this list into a table specifying which miRNAs play what role in which plants.
Lines 72-74: This is again a very heavy subject: "the involvement of miRNAs in regulating the embryogenic response in gymnosperms, mono- and dicot plants including economically important species"
Lines 232-243: This sentence does not make sense to me. Please try to rephrase it.
Line 237: It seems that the abbreviation SAM is only used here and is probably unnecessary.
Lines 327-330: What does "and relevantly, to" mean here? "that is" should probably be removed.
Line 464: "that inferred that similar to" does not make sense.
Line 486: the abbreviation ZE appears only once here.
Author Response
Manuscript ID: ijms-725546
Dear Editor, Lynne Liu ,
Please, find attached the revised version of the manuscript on Epigenetic regulation of auxin-induced somatic embryogenesis in plants by Barbara Wójcikowska, Anna M. Wójcik and Małgorzata D. Gaj in which all of the comments and remarks raised by the reviewer were considered.
We highlighted in yellow all of the changes made in the manuscript (relative to the previous version).
Please, find below the detailed responses to the reviewer.
Looking forward to your positive reply.
Yours sincerely,
Małgorzata Gaj
Reviewer 1
This review by Wójcikowska et al. nicely summarizes the regulatory roles of miRNAs and chemical modifications in auxin-induced somatic embryogenesis in plants. It seems that most important studies in this field were covered, but the way the review was written and presented can still be improved. I strongly suggest that the authors ask one or two biologists who are native speakers to read through the text carefully and to help improve the sentence structures and the readability of the manuscript.
My general feeling is that many sentences are too long and quite wordy, sometimes with redundant words, superfluous phrases, syntactical errors, and overuse or incorrect use of clauses/parentheticals/commas that can disrupt sentence flow and result in nonsensical parts. It should be noted that some phrases/words occur multiple times, have relatively little meaning, and often make sentences more difficult to comprehend, such as "in particular" (6 times in the text) and "relevant"/"relevantly" (10 times). Below are some suggestions:
Response:
We have improved the language in the manuscript following the Reviewer’s comments; the overused words have been reduced or replaced (e.g.. in particular, relevant, that), the long sentences with “heavy subjects” have been re-written and made shorter to improve understanding and sentence flow. All changes are highlighted.
Lines 10-11: In this example, "that accompany" can be changed to "during", which would reduce the number of "that" relative clauses in the sentence.
„The significant modulation of the gene transcription profiles that accompany SE induction results from the epigenetic factors that fine tune the gene expression towards embryogenic development.”
Response:
The sentence has been modified (l.10-13).
Line 12-15:
- Is "among these factors," necessary?
- What does "significantly and relevantly," mean?
Response:
The sentence has been modified (l.13-15).
- There are two relative clauses: "that have an ..." and "that control SE". Could they be reduced?
Response:
We have reduced the number of relative clauses in the review.
- The second part of this sentence has a very heavy subject "several miRNAs that have an essential function in regulating the transcription factors (TFs) that control SE" and a short body "have been identified in the past few years". I would rather have a shorter subject and see the finite verb earlier.
“Among these factors, microRNA molecules (miRNAs) contribute to the posttranscriptional regulation of the gene expression significantly and relevantly, several miRNAs that have an essential function in regulating the transcription factors (TFs) that control SE have been identified in the past few years.”
Response:
The sentence has been modified (l.13-15).
Line 15-17: The "Relevant ...," part doesn't seem to add any information here. The sentence says "auxin-related" twice and "auxin" thrice, which is quite redundant. I would suggest to change this sentence to "Most of these miRNAs were found to be involved in auxin metabolism and signaling."
Response:
The sentence has been modified (l.13-15).
Lines 67-72: It would be better to make this list into a table specifying which miRNAs play what role in which plants.
In addition to the model plant Arabidopsis, numerous miRNAs that have a differential expression have been identified in embryogenic cultures of a variety of other plants including Oryza sativa [14], Citrus sinensis [15, 16, 17], Liriodendron tulipifera×L. chinense [18], Dimocarpus longan [19, 20, 21], Gossypium hirsutum [22], Zea mays [23, 24, 25], Triticum aestivum [26], Larix leptolepis [27], Lilium pumilum [28], Picea balfouriana [29, 30], Cocos nucifera [31] and Pinus pinaster [32].
Response:
Following the Reviewer’s suggestion, table 1 specifying miRNAs involved in SE in different plants has been added to the main text (in the previous version of MS this table was supplementary) (l.173-175).
Lines 72-74: This is again a very heavy subject: "the involvement of miRNAs in regulating the embryogenic response in gymnosperms, mono- and dicot plants including economically important species"
“Thus, the involvement of miRNAs in regulating the embryogenic response in gymnosperms, mono- and dicot plants including economically important species has become evident (reviewed in [33], [34]).”
Response:
The sentence has been modified (l.138-144).
Lines 232-243: This sentence does not make sense to me. Please try to rephrase it.
Evidence for a link between DNA methylation and auxin signaling causes a significantly modified embryogenic response in Arabidopsis mutants that are impaired in methylases including met1 and drm [84]. A mutation in the MET1 gene, which significantly decreased the global level of DNA methylation [87], was found to promote faster in vitro shoot regeneration due to the early activation of auxin-induced WUSCHEL (WUS), which is a homeobox regulator of the shoot apical meristem (SAM) formation in embryos and an increased expression of ARF3, which is engaged in auxin signaling [88]. In addition, the met1 mutant displayed an abnormal auxin gradient in embryos and a deregulation of the PIN-FORMED1 (PIN1) auxin efflux carrier, which has an important role in polar auxin transport in plant development including SE [89, 90]. Similarly, DNA methylation was assumed to regulate the PIN-like proteins during somatic embryo development in Araucaria angustifolia [91].
Response:
We have modified this paragraph (l.293-306).
Line 237: It seems that the abbreviation SAM is only used here and is probably unnecessary.
Response:
We've removed the abbreviation ‘SAM’.
Lines 327-330: What does "and relevantly, to" mean here? "that is" should probably be removed.
„PRC2 represses the embryo maturation programs during the vegetative development in Arabidopsis [108] and relevantly, to the PRC2-depleted somatic cells of the different tissue that is acquired the embryogenic competence in response to external hormonal and stress treatments [115].”
Response:
We have changed the sentence (l. 365-370).
Line 464: "that inferred that similar to" does not make sense.
“Consistent with this model, some genetic convergence of the TSA- and 2,4-D-induced embryogenic development was revealed in Arabidopsis [49] that inferred that similar to in planta development, auxins might promote SE-regulatory genes via the histone acetylation-mediated de-repression of gene transcription.”
Response:
We've improved the sentence(l.481-484).
Line 486: the abbreviation ZE appears only once here.
Response:
It has been withdrawn.
Reviewer 2 Report
The present review ''Epigenetic regulation of auxin-induced somatic
embryogenesis in plants'' describes the Somatic embryogenesis and its complex interactions with the epigenetic factors. I think the current manuscript more focused on the role of epigenetic factors during somatic embryogenesis than auxin- epigenetics mechanism. Here are some points need to consider
- Introduction- Need to increase. Adding the background of about role of auxin in SE and epigenetic modifications will make this paper more illustrative. I think figure 1 or 1 A is better suited in the introduction.
- It is better to make extract the data from section 3-6 to prepare table/s for a better representation
- Do not find any supplementary file
- Why same title from 3-6
The current manuscript presents new information but the way the information is packaged and presented probably needs to be changed to help accentuate the important points and require a more mature presentation in a nice flow.
Author Response
The present review ''Epigenetic regulation of auxin-induced somatic embryogenesis in plants'' describes the Somatic embryogenesis and its complex interactions with the epigenetic factors. I think the current manuscript more focused on the role of epigenetic factors during somatic embryogenesis than auxin- epigenetics mechanism. Here are some points need to consider
1. Introduction- Need to increase. Adding the background of about role of auxin in SE and epigenetic modifications will make this paper more illustrative. I think figure 1 or 1 A is better suited in the introduction.
Response:
We have modified the Introduction extensively, the role of auxin in somatic embryogenesis has been emphasized and the basic epigenetic processes controlling SEinduction have been overviewed and illustrated by the newly added Figure 1. (l.27-136).
2. It is better to make extract the data from section 3-6 to prepare table/s for a better representation
DNA methylation, histone methylation, histone acetylation
Response:
General information about specific chromatin modifications and miRNAs involved in gene regulation has been extracted from different MS paragraphs and transferred to the Introduction. Figure 1 has been added to the Introduction that clearly presents all of the epigenetic processes described in the review.
3.Do not find any supplementary file
Response:
A table indicating the involvement of miRNAs in somatic embryogenesis has been transferred to the main text. (l173-175).
4. Why same title from 3-6
Response:
We have removed the repeated chapter names.
The current manuscript presents new information but the way the information is packaged and presented probably needs to be changed to help accentuate the important points and require a more mature presentation in a nice flow.
Response:
Following the reviewers' remarks, we have extensively modified the review. The sentences are now shorter, without unnecessary phrases. We hope that current version of the review is more clear, comprehensive, and has a better sentence flow.
Reviewer 3 Report
This is a well written review summarizing the complex interactions between the regulatory genes and the epigenetic effects on auxin induced somatic embryogenesis in plants.
Author Response
Reviewer 3
This is a well written review summarizing the complex interactions between the regulatory genes and the epigenetic effects on auxin induced somatic embryogenesis in plants.
Response:
We are glad to hear a good opinion about our work.
Round 2
Reviewer 2 Report
This manuscript needs careful re-reading by the authors themselves could avoid long sentences and unnecessary words. Sentences like lines- 306-309; 277-283; etc need to be defragmented.